

# Coupled influence of precipitation regimes and seedling emergence time on the reproductive strategy in *Chloris virgata*

Ying Wang[1], Jiawei Chen[1], Yige Huang[2], Zhongsheng Mu[3] and Changfu Wang[1]

[1] Jilin Jianzhu University, Key Laboratory of Songliao Aquatic Environment, Ministry of Education, Changchun, Jilin, China
[2] Environmental Geology, The Fifth Geological Survey Institute of Jilin Province, Changchun, Jilin, China
[3] Jilin Academy of Agricultural Sciences, Changchun, Jilin, China

Corresponding authors
Ying Wang,
wangying19879@aliyun.com
Changfu Wang,
wangcf1973@163.com

## ABSTRACT

Precipitation regime and seedling emergence time both influence plant growth and reproduction. However, little attention has been given to the effects of these combined factors on the reproductive strategy of *Chloris virgata*, which is a vital species in Songnen grassland. Here, we simulated the changes in the precipitation regime and seedling emergence time to evaluate tiller traits and seed production. The results showed that tiller number behaved similarly among three precipitation regimes when sowed on 15 May ($T_1$), while it increased significantly with precipitation regimes when sowed on 15 June ($T_2$) and 15 July ($T_3$). Tiller number decreased significantly with the seedling emergence time under the same water supply treatment. The proportional allocation of reproductive tiller number to total tiller number was significantly higher at $T_3$ than at $T_1$ and $T_2$. Seed number remained similar under different precipitation regimes at $T_2$ and $T_3$, whereas it was significantly lower under low precipitation than under other water levels at $T_1$. Seed number reached the maximum values at $T_2$ under the same level of precipitation treatment. Seed size was significantly lower under low precipitation compared to other water supply treatments and the lowest values in seed size, about 0.5 mg, occurred at $T_2$ under all the precipitation regimes. The lowest values in spike number were under low precipitation at all seedling emergence times. Seed yield exhibited similar trends with seed size under different precipitation regimes, while the greatest gains in these values were at $T_1$ under all the precipitation regimes. Our findings showed that simulated precipitation regimes and seedling emergence time affected the reproductive strategy of *C. virgata*. Typical and high precipitation, as well as early seedling emergence, will improve the seed yield and seed quality in this species.

## INTRODUCTION

Reproduction is one of the most important stages in a plant's life history and is the result of natural selection. Plants exhibit different reproductive strategies in response to varied environments (*Suonan et al., 2017*). Therefore, reproductive strategy plays a major role in

plant adaptation by maximizing the overall fitness of plants growing with variable resources (*Bazzaz et al., 1987*).

Seed yield, one of the vital components in plant restoration, can be determined by inflorescence number, seed number per inflorescence and seed size (*Hebblethwaite, Wright & Noble, 1980*; *Wang et al., 2010*). Therefore, environmental conditions experienced by individual plants that affect seed yield will ultimately influence vegetation restoration. Another vital factor for determining seedling establishment and regeneration is seed biology, which is thought to be an important contributor in understanding plant ecological processes within communities (*Venable & Brown, 1988*). Seed biology is mainly affected by seed genotype and parental environment (*He et al., 2014*). Maternal nutrition has been demonstrated as a vital factor that impacts the competitive ability of offspring (*Galloway, 2001*). Selection pressure for increased seed size may also arise because larger seeds are better at surviving hazards of seedling establishment such as nutrient shortages (*Metcalfe & Grubb, 1997*), water availability (*Bidgoly et al., 2018*; *Müller et al., 2019*) and temperature (*Dewan et al., 2018*). Consequently, studies of the reproductive strategies of plants are useful in understanding the adaptation of plants to their natural environment.

Degradation has been severe in the Songnen grassland of China including bare patches that have appeared in this region due to soil salinity and alkalinity. *Chloris virgata* Sw. is an annual tuft grass with substantial drought and alkali tolerance. Moreover, it is the first species to form a relatively stable and productive single-species dominant community (*Zheng & Li, 1999*) and is therefore a vital species to restore degraded semi-arid grassland.

Precipitation is one of the major factors that limits seed productivity in arid and semiarid areas. Inevitably, the precipitation regime may profoundly influence reproductive traits in perennial and annual grasses (*Wang et al., 2010*; *Wang et al., 2018*). Moreover, soil water content has been used to successfully predict seedling emergence (*Ikeda et al., 2019*). Spring precipitation effects on the vegetation spring phenology in arid and semi-arid regions (*Shen et al., 2015*). Consequently, there can be major impacts on seedling emergence time due to early or late spring precipitation in wet or dry years, with seedling emergence time influencing the plant's fitness within its community (*Du & Huang, 2008*).

Therefore, the objective of this study was to investigate the reproductive traits of this important species in response to combined effects of precipitation regime and seedling emergence time. The findings on the reproductive ecology of this species will inform rational regulatory and management policy for grassland restoration.

## METHODS AND MATERIALS

### Experimental design and treatment

The experiment was conducted at Jilin Jianzhu University, Jilin Province, China in 2016. *C. virgata* seeds were obtained from the Greenhouse of Jilin Jianzhu University, Jilin Province, China (44°33N, 123°82 31E) in 2015. Twenty seven PVC pipes, 25 cm diameter and 50 cm height, were filled with saline-sodic soils at different sowing dates in 2016. The soil was collected from the surface layer (0–20 cm) of a grassland near the Northeast Normal University field experiment station, in Changling, Jilin Province, China (123°44

0E, 44°44 0N, 167 m above sea level). The soil total nitrogen was 6.9%, organic carbon 0.4%, pH 8.6, electrical conductivity 91 μs cm$^{-1}$ and field capacity of 200 g kg$^{-1}$. And the maximum/minimum temperatures were 32/16 °C (day/night) during the treatment period from 15 June to 15 September. The soil-filled pipes were all buried in the soil and the top edge of each pipe and the soil surface were kept at the same level. Twenty seeds were sown on 15 May ($T_1$), 15 June ($T_2$) and 15 July ($T_3$), respectively, to ensure three seedling emergence times (SETs). Most seedlings emerged 5 d after sowing and 10 uniform and robust seedlings were selected per pipe 10 d after sowing. Three treatments were used to simulate local precipitation regimes (PRs), corresponding to 120 (L: low, soil moisture was 6.0%), 200 (T: typical, soil moisture was 11.0%), and 280 (H: high, soil moisture was 19.0%) mm precipitation, and these were created by adding water every 5 d to the pipes from 30 d after seedling emergence (see *Wang et al., 2018*). The typical precipitation regime was based on the average amount and frequency of the local region's precipitation (June–October) over the past 21 years in the Songnen grassland, where *C. virgata* is widely distributed. The low and high precipitation regimes were 40% lower and higher than the typical precipitation regime, based on data from the Songnen grassland in wet and dry years, respectively. For example, the plants were watered 12.5 mm 5 d intervals (The number of precipitation was 16 from 15 June to 15 September because most precipitations occurred in this period) under typical precipitation regime. Then the precipitation was calculated according to the precipitation depth and cross section of the pipe. There were 3 replicates for each precipitation level at each seedling emergence time, giving 30 individuals per treatment for determining tiller traits and seed production. There was a total of 27 pipes across all treatments for the three seedling emergence times and the pipes were arranged outdoors in a replicated randomized block design under a rain shelter made from transparent 0.2 mm polyethylene sheeting on 15th May, 2016.

## Data collection

The plants from each treatment at each seedling emergence time were harvested in mid-September, when all *C. virgata* individuals were fully mature. Some pipes were damaged by locusts during the course of the experiment and were excluded from the analyses, giving 22–30 individuals. The numbers of reproductive and vegetative tillers of *C. virgata* were recorded for each individual in each pipe. One first-order tiller was selected randomly from each individual in each pipe. Then the numbers of mature seeds and unripe seeds were counted to obtain seed number. The number of left spikes from each plant were recorded and the spikes were kept in individual paper bags. Fifty seeds were selected randomly from each plant and dried at 65 °C for 48 h and weighed to determine mean seed size. The remaining seeds from each plant were dried at 65 °C for 48 h and weighed to obtain seed yield.

## Data analysis

All data were tested for the normality and homogeneity and subjected to analyze by SPSS statistical software (version 17.0, SPSS Inc., Chicago, IL). One-way ANOVA was used to analyze variables such as tiller number, proportion of reproductive tiller number to total

**Table 1  Analysis of variance of vegetative tiller number, reproductive tiller number, seed number, seed size, spike number and seed yield of *C. virgata*.**

| Source | Df | Mean Square | F | Sig. |
|---|---|---|---|---|
| Vegetative tiller number (n plant$^{-1}$) | | | | |
| SET | 2 | 165.964 | 83.377 | .000[***] |
| PR | 2 | 0.078 | 0.039 | .962 |
| SET * PR | 4 | 3.402 | 1.709 | .148 |
| Reproductive tiller number (n plant$^{-1}$) | | | | |
| SET | 2 | 16.592 | 11.963 | .000[***] |
| PR | 2 | 18.896 | 13.625 | .001[***] |
| SET * PR | 4 | 1.741 | 1.255 | .288 |
| Seed number (n spike$^{-1}$) | | | | |
| SET | 2 | 732947.466 | 41.028 | .000[***] |
| PR | 2 | 99431.410 | 5.566 | .004[**] |
| SET * PR | 4 | 22621.282 | 1.266 | .284 |
| Seed size (mg) | | | | |
| SET | 2 | 0.313 | 38.160 | .000[***] |
| PR | 2 | 0.134 | 16.318 | .000[***] |
| SET * PR | 4 | 0.020 | 2.498 | .043[*] |
| Spike number (n plant$^{-1}$) | | | | |
| SET | 2 | 72.017 | 6.715 | .001[***] |
| PR | 2 | 182.704 | 17.036 | .000[***] |
| SET * PR | 4 | 13.148 | 1.226 | .300 |
| Seed yield (g m$^{-2}$) | | | | |
| SET | 2 | 61.511 | 36.073 | .000[***] |
| PR | 2 | 39.863 | 23.378 | .000[***] |
| SET * PR | 4 | 6.003 | 3.52 | .027[*] |

**Notes.**
[*]Significant at the 0.05 statistical level.
[**]Significant at the 0.01 statistical level.
[***]Significant at the 0.001 statistical level.

tiller number, seed number, spike number, seed size and seed yield. Two-way ANOVA was also used to analyze all data under different precipitation regimes over seedling emergence times. Differences under different precipitation regimes and seedling emergence times were analyzed using the least significant difference (l.s.d.) test.

## RESULTS

### Vegetative and reproductive tiller growth

The ANOVA results showed that seedling emergence time individually had a significant effect on vegetative tiller number, and that precipitation regime and seedling emergence time individually had significant effects on reproductive tiller number. However, their interactions had no effect on vegetative tiller number or reproductive tiller number (Table 1).

Growth and reproduction in the *C. virgata* plants responded differently to different levels of precipitation and seedling emergence times (Fig. 1). The vegetative tiller number

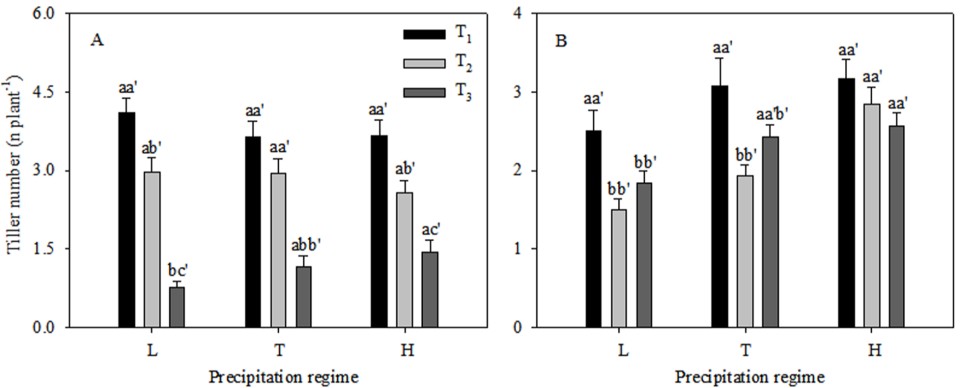

**Figure 1** **Vegetative tiller number (A) and reproductive tiller number (B) (mean ± 1 s.e.) of *C. virgata* under different precipitation regimes at different seedling emergence times.** Letters a, b, and c are used for comparisons between different precipitation regimes, while a′, b′ and c′ are used for comparisons between different seedling emergence times. Means with the same letter are not significantly different at $P = 0.05$.

was similar under different precipitation regimes at $T_1$ and $T_2$, while significant variations were found among the three precipitation regimes at $T_3$ (Fig. 1A). The vegetative tiller number varied significantly within increasing precipitation and the vegetative tiller number was significantly higher under high precipitation than under low precipitation at $T_3$. There was no significant change in reproductive tiller number with increasing precipitation at $T_1$ (Fig. 1B). At $T_2$ the reproductive tiller number was similar between low and typical precipitation but was significantly higher under high precipitation. At $T_3$ the reproductive tiller number was similar under typical and high precipitation treatments and was significantly higher than under low precipitation. Vegetative tiller number decreased significantly from $T_1$ to $T_3$ under all the precipitation regimes. The highest reproductive tiller numbers occurred at $T_1$ under the low precipitation regime. The reproductive tiller numbers were significantly higher at $T_1$ than at $T_2$ under typical precipitation. Furthermore, changes in seedling emergence time did not affect reproductive tiller number under the high precipitation regime (Fig. 1). The proportion of reproductive tiller number to total tiller number were similar between $T_1$ and $T_2$, while were significantly lower than that at $T_3$ (Fig. 2).

## Seed production

The ANOVA results showed that precipitation regime and seedling emergence time individually had significant effects on seed number, seed size, spike number and seed yield (Table 1). Their interactions had an effect on seed size and seed yield. However, their interactions had no effect on seed number and spike number (Table 1).

The seed number and seed size of the *C. virgata* plants were significantly different under the various precipitation regimes and seedling emergence times (Fig. 3). Seed number remained unchanged among the three precipitation regimes at $T_2$ and $T_3$, whereas seed number was significantly lower under low precipitation than under the typical and high precipitation regimes at $T_1$. Within the same precipitation treatment seed numbers were

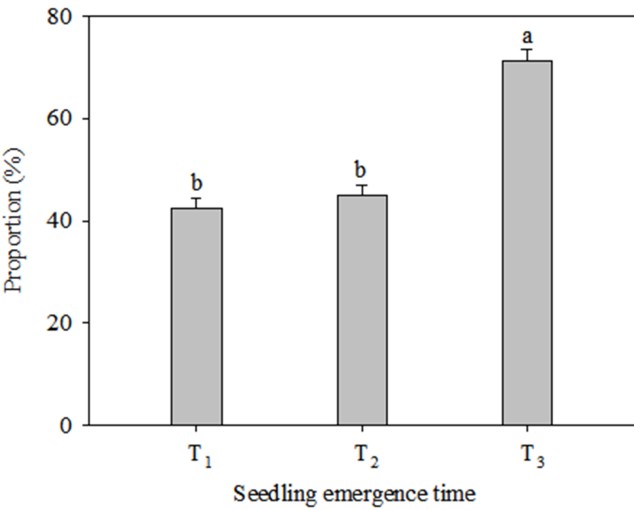

**Figure 2  Proportion of reproductive tiller number to total tiller number (mean ± 1 s.e.) of *C. virgata* at different seedling emergence times.** A, b, and c are used for comparisons between different precipitation regimes. Means with the same letter are not significantly different at $P = 0.05$.

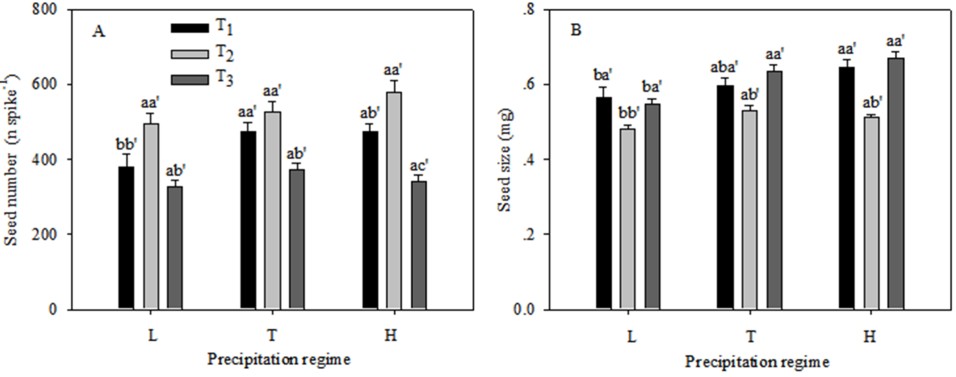

**Figure 3  Seed number (A) and seed size (B) (mean ± 1 s.e.) of *C. virgata* under different precipitation regimes at different sowing times.** A, b, and c are used for comparisons between different precipitation regimes, while a′, b′ and c′ are used for comparisons between different seedling emergence times. Means with the same letter are not significantly different at $P = 0.05$.

similar at $T_1$ and $T_2$ and were significantly higher than at $T_3$ (Fig. 3A). Seed size was unchanged from $T_1$ to $T_3$ under typical and high precipitation and was higher than under low precipitation at all sowing times. Within the same precipitation treatment seed size remained unchanged at $T_1$ and $T_3$ and was higher than at $T_2$ (Fig. 3B).

There was a significant difference in spike number among different levels of precipitation at different seedling emergence times (Table 2). Spike number increased from the low to typical precipitation regimes but under high precipitation there was no further significant change at $T_1$. There was a significant difference between low and high precipitation at $T_2$. Spike numbers increased significantly from low to high precipitation at $T_3$. Spike
**Table 2  Spike number (mean ± 1 s.e.) of *C. virgata* under different precipitation regimes at different seedling emergence times.**

|   | $T_1$ | $T_2$ | $T_3$ |
|---|---|---|---|
| L | 3.5 ± 0.41[bA] | 2.5 ± 0.33[bA] | 2.6 ± 0.32[cA] |
| T | 6.5 ± 1.07[aA] | 3.5 ± 0.37[abB] | 4.6 ± 0.61[bAB] |
| H | 6.1 ± 0.86[aA] | 4.6 ± 0.40[aA] | 6.2 ± 0.55[aA] |

**Notes.**
Within a column, means followed by the same lower case letter are not significantly different at $P = 0.05$.
Within a line, means followed by the same capital letter are not significantly different at $P = 0.05$.

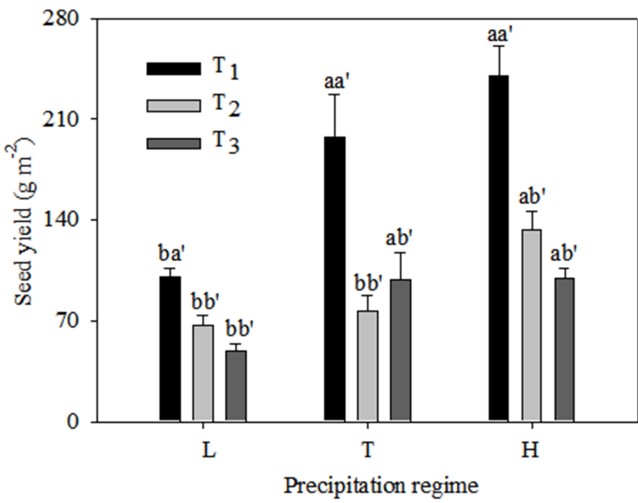

**Figure 4  Seed yield (mean ± 1 s.e.) of *C. virgata* under different precipitation regimes at different seedling emergence times.** A, b, and c are used for comparisons between different precipitation regimes, while a′, b′ and c′ are used for comparisons between different seedling emergence times. Means with the same letter are not significantly different at $P = 0.05$.

numbers remained unchanged from $T_1$ to $T_3$ under low and high precipitation. While the spike numbers under the typical precipitation regime decreased substantially from $T_1$ to $T_2$. There was no significant difference between $T_1$ and $T_3$, or $T_2$ and $T_3$ under typical precipitation.

Seed yield was significantly higher under the typical precipitation regime than under low precipitation at $T_1$ and $T_3$, but there was no difference between typical and high precipitation for these two emergence times (Fig. 4). At $T_2$ there was no significant difference in seed yield between low and typical precipitation but there was a marked increase under high precipitation. The highest yield values occurred at $T_1$ under all precipitation regimes.

## DISCUSSION

### Vegetative and reproductive tiller growth
Plants select different reproductive strategies according to their growing environment, and the reproductive strategy selected in a given environment should maximize fitness (*Zhang et*

*al., 2018*). However, resources are the primary factor determining how plants establish their reproductive machinery (*Weiner et al., 2009*). Reproductive tiller number is an important characteristic influencing seed yield in many plants (*Peng et al., 2015*). Whether a bud develops into a reproductive or a vegetative tiller is based on the environment and the evolutionary history of the population (*Warringa & Kreuzer, 1996*). Our results showed that the trends in tiller number in *C. virgata* under different precipitation regimes were similar at $T_1$ and $T_2$ but were different at $T_3$. The trends were similar with the results in 2018 when sowed at the early date. It can be concluded that water might be one of the limiting resource for tiller number at $T_3$. Furthermore, the proportional allocation of reproductive tiller number to total tiller number was significantly higher at $T_3$ than at $T_1$ and $T_2$ (Fig. 2). High temperature improved the proportion of reproductive tiller number to total tiller number and the results were consistent with those from previous study (*Escalada & Plucknett, 1975*). Plant morphology was affected by temperature and daylength, while many plants did not react to critical periods of photoperiod unless their thermal requirements were met. *C. virgata* produced tiller 10 days later after sowing. The temperature was higher when *C. virgata* produced tillers at $T_3$ than that at $T_1$ and $T_2$. Delayed sowing and low precipitation reduced *C. virgata* growth (total tiller number), while maximized it's reproduction and restoration potential. This strategy has also evolved in monocarpic species and according to optimal allocation theory, such plants should allocate more resources to reproductive parts to improve seed production in unfavorable environments (*Ellner, 1987*).

## Seed production

Seed production are related to plant adaptations to their growing environment. For example, the maternal nutrient environment affects seed size, seed yield and the competitive ability of progeny (*Gómez, 2004*). In our study, seed size was smaller at $T_2$ compared to the other two emergence times. The larger seed size at $T_1$ and $T_3$ may indicate higher seed quality because large seed size has been associated with higher seed germination ability and seedling survival as reported by other authors (*Cowell & Doyle, 1993*) and in our earlier work (*Wang et al., 2018*). This also suggests a management method to produce high quality seeds from *C. virgata*.

Moreover, our results showed that when *C. virgata* was delayed in emergence beyond 30 d or 60 d it produced fewer seeds. It has been demonstrated that sowing date was one of the critical factors for productivity of winter wheat (*Triticum aestivum* L.) due to different amounts of soil water storage at different sowing times (*Ren et al., 2019*). Further, *Awan & Chauhan (2016)* have reported that a delay in emergence would result in fewer seeds returning to the seed bank, which has important implications for vegetation succession and restoration (*Shang et al., 2016*). However, seed yields were similar among the three water supply treatments and the first-order spike number was one of the most important factors in determining seed yield in our former work (*Wang et al., 2018*). Seed yields in the current work were higher in typical and high precipitation than under low precipitation. The pattern of seed yield was not consistent with our former work in 2018, and the reason for this might be the different plant densities in the two experiments. However, the first-order

spike number (reproductive tiller number) was higher under low precipitation than under typical and high precipitation in these earlier experiments under a density of five plants per pot, whereas the first-order spike number was significantly higher under typical and high precipitation compared to low precipitation with a density of ten plants per pot in the current work (Fig. 1). This indicated the existence of different reproductive strategies under different plant densities. Here, reproductive tiller number was demonstrated to be an important component for *C. virgata* seed yield, and future work should explore the effect of plant densities on growth and reproductive characteristics.

## CONCLUSION

In conclusion, simulated precipitation regimes and seedling emergence time affect the reproductive strategy of *C. virgata*. Increased precipitation did increase seed yield in this study, which suggested that reproduction in *C. virgata* benefits from average and above-average precipitation regimes. In addition, dry years or delayed precipitation during seedling emergence due to climate change might have a negative effect on plant recovery of *C. virgata* in arid and semiarid areas such as the Songnen grassland. Early sowing in spring might be a strategy to ensure high seed yield and high seed quality by avoiding late-season drought conditions for this species. These results will also provide some important information on management measures for the establishment and recovery of *C. virgata* in degraded grassland.

### Funding
This work was financed by the National Keypoint Research and Invention Program (2019YFD1002603) and the National Nature Science Foundation of China (51678273). The funders had no role in study design, data collection and analysis, decision to publish, or preparation of the manuscript.

### Grant Disclosures
The following grant information was disclosed by the authors:
National Keypoint Research and Invention Program: 2019YFD1002603.
National Nature Science Foundation of China: 51678273.

### Competing Interests
The authors declare there are no competing interests.

### Author Contributions
- Ying Wang conceived and designed the experiments, prepared figures and/or tables, authored or reviewed drafts of the paper, and approved the final draft.
- Jiawei Chen performed the experiments, prepared figures and/or tables, and approved the final draft.

 

- Yige Huang and Zhongsheng Mu performed the experiments, analyzed the data, prepared figures and/or tables, and approved the final draft.
- Changfu Wang conceived and designed the experiments, authored or reviewed drafts of the paper, and approved the final draft.

### Data Availability

The raw data is available in Supplemental File.

### Supplemental Information

Supplemental information for this article can be found online at http://dx.doi.org/10.7717/peerj.8476#supplemental-information.

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
