# Peer review of "Coupled influence of precipitation regimes and seedling emergence time on the reproductive strategy in Chloris virgata"

_PeerJ, doi:10.7717/peerj.8476_

## Round 0.1 · original submission · Major Revisions

Dear Dr. Wang

Thank you for submitting your manuscript to PeerJ. The reviewers and I have made suggestions which I feel would improve your manuscript. I encourage you to consider these comments and make an appropriate revision of your manuscript. The comments are below.

Please check the manuscript carefully for a concise flow, referencing and English language.

Some information in M&M section is missing. Please provide information about the principal atmospheric and soil variables along the experimental period, experiment duration, soil physicochemical properties. A better justification of each specific treatment level should be done.

I am not satisfied with the statistical model used. Many of the recorded parameters are discrete in nature, such as tiller and seed number, by definition Anovas model are not appropriated for these kinds of results without its transformation in order to assure its adjustment to a normal model. Was normality and homogeneity of the results tested?. Authors should apply other more robust two factors statistical model, maybe GLM model, which contribute to explain the influence of both main factors tested (emergence time and precipitation regime) and its interactive effect in each specific parameter. I suggest a single Table giving the main effects and interactions for all parameters. Considering this aspect the results section should be redrafted considering mainly the interaction effect since this is the whole point of study. Also statistical results should be used more consistently through the text.
Why not other growth parameters were used, such tiller height and density, above and belowground biomass?. This information could be useful for specie which intended to be used for restoration purpose together with the reproductive traits.

Please check the legends on the figures and/or tables for completeness, such as if values are means, SD or SE, n = replicates, and also the details and explanations of any axis and statistical labels and the name of the test used and p value. Letters used in the figures should be fitted to a two factor model, considering the interaction between both experimental factors.

There is a lack of ecological explanation through the discussion section, especially with Non-reproductive and reproductive tiller results. Please treat to provide a mechanistic explanation for the obtained results and in what extent this could affects the potential of this specie for restoration or management tasks. In addition, please improve the comparison with the results published by the authors in 2018.

Reviewer 1 ·

Basic reporting

1.Materials and methods are acceptable. My question is why do you chose this species again? You have done the work about the effects of precipitation regime on reproductive strategy of this species in the former work. Why did you not design the coupled factors in hte former work?
2.Figure 3, the legend (T1, T2 and T3) should appear in Figure 3A not Figure 3B.
3. Line 72 and 91, the blank spaces should be deleted.
4.Line 69 The sentence: “Precipitation is one of the major factors that limits seed productivity in arid and semiarid areas”. Did author research on other factors, for example nitrogen, temperature or extreme weather? And which factor is more important for this species in the research?
5. Line 106: The left spikes from each plant were recorded and kept in individual paper bags. The authors meant that "the number of left spikes from each plant were recorded and the spikes were kept in individual paper bags"?
6. Seed yield in Figure 3B. Why the data is so different with the former study (in 2018)?
7. Please check the format of the journal and revise the references as the journal requirement. In addition, you can delete some older ones if they are not so important.  
8. The “Spike number (plant-1)” in Table 3 should be“Spike number (n plant-1)” ?
9. Please check the format of the journal and revised the subtitle in the manuscript as the journal requirement.
10. please check all the figure legends and revise them according to the journal requirement.

Experimental design

The experimental design is rational. Materials and methods are acceptable. The species is so important in arid and semiarid area.

Validity of the findings

There are some works about effects of precipitaion on plant growth and reproduction. While the authors can simulate the precipitation regime and precipitation time, which are important factors in the North of China. However, there was few work about the effects of two factors on plant's reproductive strategy. Therefore, it is innovative.
The raw data have been provited. The authors discussed the results linked the data. Finally, they sumarrized the findings in manuscript.

Additional comments

Comments to the Author(s)
The authors showed the reproductive strategy in Chloris virgata under coupled factors of different precipitation regimes and seedling emergence time. C. virgata is so important in recovering bare patches of alkaline soil in Songnen plain. Overall, I think the aim of the paper is interesting and worth pursuing. Especially, delayed spring precipitation occured frequently in some area.
However, there are still some flaws need further work and clarification. I suggest this paper can be published after minor revision.

Reviewer 2 ·

Basic reporting

In summary, the language, experimental design, and results of manuscript is fine. However, some section need to carefully revise. Detail informations are seen general comments for author.

Experimental design

This is a water control experiment, but I cannot find that how the author controls soil moisture. Need detail information in materials and methods section. Furthermore, the author should list the value of soils moisture in each treatment using figure or table.

Validity of the findings

no comment

Additional comments

The research collected many data and analyzed impacts of interaction of precipitation regime and seedling emergence on reproductive strategy of Chloris virgate, a vital species in Songnen grassland, China. The research providing some new support for regeneration in this annual species. However, the paper need to do some revision before acceptance.

Firstly, this is a water control experiment, but I cannot find that how the author controls soil moisture. Need detail information in materials and methods section. Furthermore, the author should list the value of soils moisture in each treatment using figure or table.

Secondly, the author using seed size and seed mass per spike in the manuscript, I think, both of them are same thing. Keep one parameter is enough. And, this is a pot experiment, but the author expressed seed yield using g per square meter. Normally, you can calculate seed yield in the field experiment, not for pot experiment. I suggest you change seed yield as seed number per pot. Thus, in the results, the author only keep spike number, seed no. per spike, seed weight per spike, seed no. per pot is enough.


Specific comments
1. Abstract: do not use T1, T2, T3 in here.
2. Results: move ANOVA results to the first paragraph at each sub-section.
3. Line 64-67. Does this species is a tuft grass? How use this species used to restore degraded grassland? Normally, it is the first species in natural succession process, but not an ideal species to plant artificial grassland. Need more explains?
4. The significance level was set at P=0.05. No need presented here. Delete.
5. Try to combine table 1, 2, 4 into one new table. Too many tables.

---

## Round 0.2 · Minor Revisions

Dear Dr. Wang

I reviewed your manuscript and I can recommended that your paper be accepted for publication in PeerJ pending minor revision (see comments below from Rev 2).

Reviewer 2 ·

Basic reporting

The authors have well revised manuscript according to the comments from reviewers and editors. However, there still needs some minor revision, particular for English expression. In summary, it should to be accept after further minor revision.

Specifically comments
1. Abstract. Line 4-5. It is not a full sentence. Need to revise. Such as: Here, we simulated the changes in the precipitation regime and seedling emergence time to clarify how the tiller and seed production response to...
2. Line 79. Revise it into Experimental design and treatment.
3. Line 105. Data collection enough.
4. Line 116. Should be Data analysis.
5. Line 123. Non-reproductive not correct. Should be Vegetative tiller.
6. Line 144. Change seed traits into Seed production.
I suggest author revise this kind of expression mistake throught the manuscript.

Experimental design

fine.

Validity of the findings

fine.

Additional comments

The authors have well revised manuscript according to the comments from reviewers and editors. However, there still needs some minor revision, particular for English expression. In summary, it should to be accept after further minor revision.

Specifically comments
1. Abstract. Line 4-5. It is not a full sentence. Need to revise. Such as: Here, we simulated the changes in the precipitation regime and seedling emergence time to clarify how the tiller and seed production response to...
2. Line 79. Revise it into Experimental design and treatment.
3. Line 105. Data collection enough.
4. Line 116. Should be Data analysis.
5. Line 123. Non-reproductive not correct. Should be Vegetative tiller.
6. Line 144. Change seed traits into Seed production.
I suggest author revise this kind of expression mistake throught the manuscript.

---

## Round 0.3 · accepted · Accept

Dear Dr Wang and Dr Mu,

A final disposition of "Accept" has been registered for the above-mentioned manuscript.

Kind regards,